# OpenReview forum: "Infini-gram: Scaling Unbounded n-gram Language Models to a Trillion Tokens"
_colmweb.org/COLM/2024/Conference — COLM_

### Official Review · Reviewer_DJ1F · 2024-05-09

**Rating:** 7
**Confidence:** 3
**Ethics Flag:** 1

**Summary:**

This paper primarily proposes improvements to the ngram language model, enhancing its capabilities through engineering optimizations that support grams of unlimited length, while also utilizing a larger-scale corpus. The ngram language model serves as an excellent complement to Large Language Models (LLMs).

**Questions To Authors:**

1. Compared to LLMs, besides calculating PPL, in what scenarios can the infinity ngram model apply?
2. How can the Ngram model complement LLMs?
3. Is it possible to directly generate text like auto regressive model?
4. How does the generalization of the Ngram model?

**Reasons To Accept:**

1. This paper studies the Infinity-ngram model, which is an interesting issue and can also be helpful for the research on Large Language Models (LLMs).
2. This paper, through engineering optimizations, supports a large corpus and "infinity" ngram length. It also provides inspiration for subsequent related research.

**Reasons To Reject:**

1. The model proposed in the paper lacks practical experiments, making it difficult to see its future prospects.

---

> ### Author Rebuttal · Authors · 2024-05-30
>
> Thanks for your feedback! We appreciate your recognition of our contributions in scaling n-gram LMs and using them to enhance neural LLMs.
>
> **[Reasons-to-Reject 1/Q1/Q3] Besides PPL, in what scenarios can infini-gram be applied to/Can infini-gram directly generate text:** We believe that ∞-gram LMs have the potential to be useful for a wide array of end tasks through text generation, similar to how LLMs are being used. In our preliminary investigation, we identified some unique challenges with ∞-gram LMs in text generation, as discussed at the end of Section 5.2, e.g., not being robust to digression, and that it may require new sampling algorithms. We defer such development to future research.
>
> **[RR1] Model lacks practical experiments and future prospects:** In addition to the ∞-gram model, our key contribution also includes the infini-gram text search engine. We opened an API endpoint of the infini-gram engine for public use, and during the past three months, it has served over **60 million queries**. It is evident that the technology we built for this paper has become a useful research tool for the community.
>
> **[Q2] How can the ngram model complement LLMs:** Our paper shows one way -- by interpolating the probabilities given by both models. This method is simple but may be limited in expressiveness. In future work, it is possible to develop more advanced ways for n-gram LMs to complement neural LMs, e.g., training neural LMs to fit the residue between n-gram LMs and the ground truth, or using n-gram LMs to validate neural LM generations.
>
> **[Q4] Generalization of the n-gram model:** The n-gram LM alone has limited generalization due to data sparsity. We explored a distribution shift scenario (time shift) in Appendix D.3 and found the gains are reduced (although remain positive). We think the n-gram LM is particularly useful when verbatim memorization is more important than generalization, e.g., entity names and common expressions; see our response on “error analysis” to Reviewer fyaK for more details. Additionally, our combined model, which interpolates the n-gram LM with the neural LM, partially addresses this issue, contributing to the positive gains in Appendix D.3.

---

> > ### Comment · Reviewer_DJ1F · 2024-06-07
> >
> > Thank you for your response. Thanks for such an interesting paper

---

### Official Review · Reviewer_fyaK · 2024-05-09

**Rating:** 7
**Confidence:** 5
**Ethics Flag:** 1

**Summary:**

This paper presents infini-gram, an n-gram model with unbounded n-gram order trained on very sized corpora containing 5T tokens. The paper shows that infini-gram can substantially outperform vanilla 5-gram models in terms of next token prediction (62% relative improvement) and can yield improvement in perplexity upon interpolating with neural lms. The paper shows that infini-gram can be implemented efficiently using suffix arrays, a technique that has been proposed in the past but not used for scaling the LM to trillions of tokens.

Strengths:
* Shows that an n-gram model can be scaled to substantially large corpora with trillions of tokens.
* Shows that an n-gram model can yield perplexity improvements when combined with neural LMs
* Demonstrates how the infini-gram can be used to conduct analysis of effective n (length of longest suffix with non-zero count) in neural LMs.

Weaknesses:
* There are some critical metrics which are not reported in the paper
e.g. next token prediction for neural LM, perplexity of the infini-gram by itself
*  Though the infini-gram outperforms a 5-gram LM, it is still substantially weaker than a neural LM. While it is good to see that the infini-gram improves the perplexity of the neural LM, this result has been reported in prior literature
https://www.microsoft.com/en-us/research/wp-content/uploads/2016/02/rnn_ctxt.pdf
* The paper does not report comparison with other non-parametric approaches such as kNN LMs (Khandelwal et al. 2020)

**Questions To Authors:**

* "When plotting the agreement level with respect to the suffix length, text generated by neural LMs with nucleus sampling is most similar to human-written text, among other decoding methods like greedy decoding and temperature sampling. For greedy decoding, the agreement plot suffers from significant fluctuation, which may be rooted in deficiencies in neural LM pretraining and the positional embeddings of Transformer"
It is unclear how this finding is related to infini-gram.
*  Figure 3: what is the next token prediction accuracy for neural LM?

**Reasons To Accept:**

* Shows that an n-gram models can be scaled to very large corpora with trillions of tokens - which are used currently for training neural LMs
* Shows that the n-gram model can be used to perform interesting text analysis of neural LMs using metrics such as 'effective n' - which is the length of the longest suffix with non-zero count in training data
* Demonstrates that the infini-gram model can be implemented efficiently using suffix arrays. Although this technique has been used in the past, this paper may further popularize its application in the research community

**Reasons To Reject:**

* The paper does not report key metrics such as next-token prediction accuracy of neural LM, the perplexity of infini-gram LM by itself.
* The paper does not compare the proposed approach to other non-parametric LM approaches such as the kNN LM (Khandelwal et al)
* The paper does not present an error analysis showing where the infini-gram either outperforms/underperforms a neural LM. e.g. is a neural LM better at generalization?

---

> ### Author Rebuttal · Authors · 2024-05-30
>
> Thanks for recognizing the values in scaling n-gram LMs and their effectiveness to performance improvements and analysis.
>
> **[Weakness 1/Reasons-to-Reject 1/Q2] Missing metrics:** We report the suggested metrics below, using the same eval data as in Sec 4.1 (“Setup”).
> * **Accuracy of next-token prediction:**
>   * Neural-only: 62%, 63%, and 65%, with Llama-2 7B, 13B, and 70B, respectively
>   * Neural + ∞-gram: 66%, 67% and 69%, with Llama-2 7B, 13B, and 70B, respectively
>   * ∞-gram only: 47%
> * Consistent with the perplexity results, ∞-gram supplements neural LMs, although using ∞-gram alone does not outperform neural LMs.
> * **Perplexity of ∞-gram LM alone:** Since ∞-gram probabilities may contain zeros, we evaluated a variant of ∞-gram that takes a weighted average of all n-gram LM predictions where n ranges from 1 to the longest suffix. **It has a perplexity of 19.82.** In comparison, GPT-2 (117M) is 22.82 and GPT-2 (345M) is 16.45 (Table 1).
> * We highlight that ∞-gram alone does not outperform neural LMs; rather, ∞-gram supplements neural LLMs, making smaller LLMs comparable to larger LLMs.
>
> **[Weakness 2] Prior literature:** We will revise our paper to highlight that the interpolation method is not new, but our key contribution is showing that ∞-gram can supplement **large** LMs (the neural model in the 2016 paper has 300 neurons, whereas we compared with neural LMs up to 70B parameters).
>
> **[Weakness 3/RR2] Comparison with kNN-LM:** We included a comparison of ∞-gram with other non-parametric methods (kNN-LM, RIC-LM) in Appendix D.2 and Table 4. We will move them to main text in the revision. ∞-gram improves perplexity much more than these alternatives. Also, as kNN-LM and RIC-LM are more resource-intensive, prior work could not scale the reference data to trillion-token scale.
>
> **[RR3] Error analysis:** We did some qualitative analyses, and found that ∞-gram is often good at completing multi-token words (e.g., hippop**otamus**, correctly predicted tokens are bolded), common phrases (e.g., born **in**), and entity names (e.g., educated at Tr**inity College**). ∞-gram is not very good at recalling factual knowledge (e.g., predicting the first token of an entity name), likely due to insufficient contextualization. We’ll include this analysis in the next revision.
>
> **[Q1] It is unclear how “When plotting ... Transformers” is related to infini-gram:** This paragraph in Sec 4 is an example of the type of insights we can gain about LLMs, using ∞-gram as a tool.

---

> ### Comment · Reviewer_fyaK · 2024-06-03
>
> Thanks to the authors for providing additional details on metrics and answering my questions. I have increased my scores.

---

### Official Review · Reviewer_CssV · 2024-05-10

**Rating:** 7
**Confidence:** 4
**Ethics Flag:** 1

**Summary:**

The paper presents an approach and method to train n-gram language models with arbitrarily large n, as well as training of n-gram models on trillion-scale datasets that match in size the datasets used for training neural language models. The approach to training infini-gram models is novel and allows training of models that perform significantly better than traditional n-gram models in next token prediction task. The authors also show that combining neural language model with an infini-gram model improves model perplexity significantly.  All in all this is a good paper. It is well structured, the approach to extending n-gram context length is to my understanding novel and the results are promising. This is also the first paper to train n-gram models with a 5 trillion token dataset.

**Reasons To Accept:**

The approach and method is novel, the results are good and the paper is well written.

**Reasons To Reject:**

It could be argued that given the availability of compute resources and the performance of large neural language models, there is no need for n-gram models, especially as they cannot match neural language models in their capabilities. However, given the huge burden on environment that training large neural language models has, researching and developing alternative and less resource intensive methods is a welcome direction.

---

> ### Author Rebuttal · Authors · 2024-05-30
>
> Thanks for recognizing the novelty of our method, which enabled training n-gram LMs at unprecedented scale.
>
> **Regarding Reasons to Reject (“There may be no need for n-gram models because compute resources for LLMs are already available and the n-gram model does not match neural LMs”):** While it is true that the ∞-gram LM alone does not outperform neural LMs, it is important to note that ∞-gram LMs can supplement neural LMs, making smaller models comparable to larger models, e.g., 7B model with ∞-gram matches 70B model, as shown in Table 1. We believe this is still a meaningful research direction, given that training, developing, and using massive-scale LLMs is still resource-heavy and is infeasible to do at most institutions.
>
> In addition, we opened an API endpoint of the infini-gram engine for public use, and during the past three months, it has served over **60 million queries**. It is evident that the technology we built for this paper has become a useful research tool for the community.

---

> > ### Comment · Reviewer_CssV · 2024-06-04
> >
> > Thank you for your response. I agree that the research direction is very relevant and that there could be use cases with the need for such models. I don’t see a reason to change the score.

---

### Official Review · Reviewer_Quuz · 2024-05-13

**Rating:** 9
**Confidence:** 5
**Ethics Flag:** 1

**Summary:**

Authors present a non-parameteric language model based on suffix trees, they extend the ngram tables in the following ways:
(1) The consider the longest prefix that has non-zero count in the training corpus (n is variable and tend to be large)
(2) The probability estimate is an interpolation between the neural LM and count based LM
(3) Training corpus is in the scale of LLMs (trillion of tokens)

The authors combine parameteric models NeuralLM with their count based \inf-gram method.

The authors propose new metrics to analyze the performance of their \inf-gram method by studying the agreement between the neural method probability and the count based method. They use this agreement metric to analyze models of different sizes and try to distinguish human writing from machine one.

**Questions To Authors:**

(1) Can you comment or possibility of extending the agreement analysis as a basis for machine detection of human writing?

(2) Would extending the training corpus of the ngrams follow a scaling law trend?


(3) I got confused by byte-pair and tokenized text representation used to fill the suffix tree, Would you count this a byte-level model? Maybe in the camera ready version try to walk the reader slowly with how the suffix tree being built using a toy example.

**Reasons To Accept:**

(1) I think the community will find this paper thought provoking: most of the recent work is on parameteric LM. Combining non-parameteric models shows improvement in our ability to maximize the likelihood of the evaluation corpus. Reviving this old approach might lead to the following interesting applications:
(a) Possible ways to improve generation with a large predefined factual corpus.
(b) Possible ways to identify (fingerprint) human-generated corpus vs machine one.



(2) The paper is well written and easy to follow with honest description of what works (improving likelihood) and what does not work yet (generation). I really enjoyed the balanced tone of the work and the focus on the analysis.

**Reasons To Reject:**

(1) I found the sections describing the suffix trees and how the tokenizer interact with the suffix trees not as clear as the rest of text.
(2) Lack of scaling law analysis with the impact of the training corpus (data) on perplexity.

---

> ### Author Rebuttal · Authors · 2024-05-30
>
> We really appreciate your recognition of our paper, especially the applications that may be enabled by the technology we develop.
>
> Here is our response to your questions:
>
> **[Reasons-to-Reject 1/Q3] Description of suffix trees and tokenizers:** We will improve the clarity of this section in our revision. This is a token-level model, not a byte-level model, because the smallest unit is a token and we don’t use raw ASCII/Unicode bytes. We first use an existing tokenizer to convert the dataset into an array of token IDs, and then build the suffix array for this token ID array. The byte-pair thing is an implementation detail: each token ID happens to occupy 2 bytes in little-endian order, so the suffix array for the token ID array is slightly different from the suffix array for the underlying bytes of this token ID array. We made sure our handling of byte order is correct and consistent.
>
> **[RR2/Q2] Scaling law analysis:** We presented this in Appendix D.4 and Figure 10 due to space limitations. Perplexity improves more with larger ∞-gram reference data, and for most domains, perplexity decreases linearly wrt the logarithm of reference data size.
>
> **[Q1] Extending the agreement analysis for machine detection of human writing:** When comparing the agreement plots in Fig 3 (middle) and Fig 5, we didn’t observe a distinguishing feature between human-written and machine-generated text. Infini-gram may become helpful in distinguishing human and machine writings, though this might require looking at other features, e.g., how “effective n” evolves over the tokens in a document. We leave this as a future research direction.

---

### Decision · Program_Chairs · 2024-07-10

**Decision:**

Accept

**Comment:**

This paper presents a nonparametric language model that generalizes traditional n-gram language models to unboundedly large n, maked scalable using suffix arrays. This makes it possible to apply the model to LLM-scale training data. While the resulting performance (understandably) lags behind neural LLMs, the paper provides quantitative evidence that an LLM can be improved by interpolating it with the proposed language model.

All four reviewers provided positive recommendations. The reviewers raised some reasonable concerns, which the authors addressed convincingly.

Overall, the paper provides an original and provocative contribution to the literature on Language Models, with convincing quantitative results.